# Health Consequences of Falls among Older Adults in India: A Systematic Review and Meta-Analysis

**DOI:** 10.3390/geriatrics8020043

**Published:** 2023-04-18

**Authors:** Isha Biswas, Busola Adebusoye, Kaushik Chattopadhyay

**Affiliations:** 1Lifespan and Population Health, School of Medicine, University of Nottingham, Nottingham NG5 1PB, UK; 2The Nottingham Centre for Evidence-Based Healthcare: A JBI Centre of Excellence, Nottingham NG5 1PB, UK

**Keywords:** falls, health consequences, older adults, India, systematic review, meta-analysis

## Abstract

Research has been conducted on the prevalence of health consequences of falls among older adults (aged ≥60 years) in India, and our systematic review and meta-analysis aimed to synthe-size the existing evidence on this topic. The JBI guideline was followed for conducting this review work. Several databases were searched, and eight studies were included. The critical appraisal scores (“yes” responses) for the included studies ranged from 56% to 78%. Among older adults in India who fell, the pooled prevalence of injuries was 65.63% (95% confidence interval [38.89, 87.96]). Similarly, head and/or neck injuries was 7.55% (4.26, 11.62), upper extremity injuries was 19.42% (16.06, 23.02), trunk injuries was 9.98% (2.01, 22.47), lower extremity injuries was 34.36% (24.07, 45.44), cuts, lacerations, abrasions, grazes, bruises and/or contusions was 37.95% (22.15, 55.16), fractures was 12.50% (7.65, 18.30), dislocations and/or sprains was 14.31% (6.03, 25.26), loss of consciousness was 5.96% (0.75, 15.08), disabilities was 10.79% (7.16, 15.02), and hospital admissions was 19.68% (15.54, 24.16). Some of the high figures indicate the need for prioritizing and addressing the problem. Furthermore, high-quality studies on this topic should be conducted, including on psychological health consequences, health-related quality of life, length of hospital stay, and death. PROSPERO registration: CRD42022332903.

## 1. Introduction

Various types of falls have been included in the International Classification of Diseases 10th Revision (ICD-10) [1], and a common one among older adults is when the person comes to rest inadvertently at a lower level [2]. Poor balance is a major contributor to falls among older adults. Balance requires the complex integration of sensory information about the body’s position relative to the surroundings and the ability to generate appropriate motor responses to control the body’s movement. The sensory component requires contribution from vision, peripheral sensation, and vestibular sense, whereas the motor component requires muscle strength, neuromuscular control, and reaction time. Linking these two components together are the higher-level neurological processes enabling anticipatory mechanisms responsible for planning a movement and adaptive mechanisms responsible for the ability to react to changing demands of a particular task. With increasing age, there is a progressive loss of functioning of sensory, motor, and central processing systems and an increased likelihood of falls. Instability and falls in older adults can be due to the impairment of any of these systems [3].

Falls can harm physical health (such as injuries and fractures), psychological health (such as depression and fear), and health-related quality of life and can even lead to death [4,5]. Social consequences include the lack of social interaction, leading to isolation [5]. Economic consequences include increased health and social care costs [5]. All these can be long-term consequences and inter-related and can ultimately take a toll on the overall quality of life [5,6].

India, a lower-middle-income country, is the second most populated country in the world, and the number of older adults aged ≥60 years is projected to increase to 198 million by 2030 and 323 million by 2050 [7]. Previous systematic reviews have reported the prevalence of falls among older adults in India to be around 26% to 37% and the pooled prevalence to be around 31% (95% confidence interval (CI) [23, 39]) [8,9]. Another systematic review reported the risk factors for falls among older adults in India and highlighted the significance of sociodemographic factors, environmental factors, health conditions, and medical interventions [10]. Research has been conducted on the health consequences of falls among older adults in India [11,12,13,14]. However, no systematic review has been conducted on this topic. Thus, this systematic review and meta-analysis aimed to synthesize the existing evidence on the prevalence of health consequences of falls among older adults in India. The findings of this review could provide a complete picture of the problem in India and help key stakeholders (including research-, policy-, and practice-related stakeholders) to prioritize the problem for addressing through the development, evaluation, and implementation of primary, secondary, and tertiary prevention interventions to prevent falls and manage its health consequences (including rehabilitation). 

## 2. Methods

The JBI guideline for prevalence systematic reviews was followed for conducting this review work [15]. The Preferred Reporting Items for Systematic Reviews and Meta-Analyses (PRISMA) guideline was used for reporting purposes [16]. The systematic review protocol was registered with PROSPERO (CRD42022332903).

### 2.1. Eligibility Criteria

Condition: Studies on the health consequences of falls were included. Studies reporting falls occurring due to intentional actions such as self-harm or domestic violence were excluded.

Context: Studies conducted in India and any setting, such as community, residential care, primary care, secondary care, and tertiary care, were included.

Population: Studies conducted among older adults (aged ≥60 years) were included. The National Policy for Older Persons was formulated in India in 1999, and it focuses on people aged 60 years and above [17]. The United Nations (UN) uses the same cut-off to define older adults [18]. If the study findings were stratified by age, data on ≥60 years were extracted. If it was not possible to extract the relevant data, the study was excluded. Studies were included if both the numerator data (i.e., the total number of health consequences of falls) and the denominator data (i.e., the total number of falls) were extractable.

Types of studies: Observational prevalence studies (e.g., cross-sectional studies) were included.

### 2.2. Search Strategy

An initial limited search was carried out on MEDLINE and Embase databases using the following keywords: “health consequences”, “falls”, and “India”. The titles and abstracts were screened for keywords, and the index terms used to describe the article were identified. The search results were inspected to ensure that relevant articles were identified. Based on this, the search strategies were developed in consultation with a Senior Research Librarian (Appendix A). The following databases were searched for published studies: MEDLINE (Ovid; since 1946), Embase (Ovid; since 1974), CINAHL (EBSCOHost; since 1945), and PsycINFO (Ovid; since 1806). ProQuest Dissertations and Theses was searched for unpublished studies. These databases were searched on 30 May 2022. No date or language restrictions were applied. The reference list of all the studies selected for inclusion in the systematic review and previous relevant reviews was screened for additional studies.

### 2.3. Study Selection

Following the searches, the identified records were collated and uploaded onto EndNote X9 (Clarivate Analytics, Philadelphia, PA, USA) [19]. After the removal of duplicate records, the titles and abstracts were screened for eligibility using the inclusion criteria by two reviewers independently (I.B. and B.A.). Studies identified as potentially eligible or those without an abstract had their full text retrieved. In case the full text of a study was unavailable even through the interlibrary loan service/British Library, the corresponding author and journal editor were approached (at least two times through email). Full texts of the studies were assessed for eligibility using the same inclusion criteria by two reviewers independently (I.B. and B.A.). Full-text studies that did not meet the inclusion criteria were excluded, and the reasons for exclusion are reported (Appendix B). During the process, any disagreements that arose between the two reviewers were resolved through discussion. If a consensus was not reached, then a third reviewer (K.C.) was involved.

### 2.4. Methodological Quality Assessment

The included studies were critically assessed independently by two reviewers (I.B. and B.A.) using the standardized JBI checklist for prevalence studies [20]. Any disagreements that arose were resolved through discussion or with the help of a third reviewer (K.C.) if a consensus was not reached. All the studies, irrespective of their methodological quality, underwent data extraction and synthesis, where possible.

### 2.5. Data Extraction

Data were extracted from the included studies using a pre-developed and pre-tested data extraction tool. Data extraction was performed independently by two reviewers (I.B. and B.A.). Any disagreements that arose between the reviewers were resolved through discussion. If a consensus was not reached, then a third reviewer (K.C.) was involved. The corresponding author of the study was contacted through email (at least two times) to request any missing data for clarification.

### 2.6. Data Synthesis

A narrative synthesis was initially conducted. Meta-analysis was conducted to estimate the pooled prevalence of health consequences of falls, along with 95% CI, using the random effects model. STATA 16 (Stata Corp LLC, College Station, TX, USA) was used for this purpose [21], and the *metaprop* command was used for this purpose [22]. The statistical heterogeneity was quantified using the *I*^2^ statistic. *I*^2^ is a percentage, and its value lies between 0% (i.e., indicates no observed heterogeneity) and 100% (i.e., indicates substantial heterogeneity). The meta-analyses of prevalence often present high *I*^2^ values. The *I*^2^ statistic is not an absolute index for the amount of variability observed, and its estimation can be impacted by factors such as the number of studies included in the meta-analysis or the pooled estimate. Similarly, a low *I*^2^ value is not always an indicator of consistent and homogenous results [23].

## 3. Results

### 3.1. Study Selection

The literature search identified 4375 records. After removing the duplicates, 3715 records were screened for eligibility. Forty-four full-text studies were screened for eligibility. Finally, eight studies were included in the systematic review and meta-analysis [11,12,13,14,24,25,26,27]. Figure 1 shows the PRISMA flow diagram of the identification, screening, and eligibility of included articles.

### 3.2. Characteristics of Included Studies

The characteristics of included studies are presented in Table 1. All the studies were cross-sectional and conducted in and after 1999 [11,12,13,14,24,25,26,27]. Five studies were conducted in the northern states of India [11,14,24,26,27], and three in southern India [12,13,25]. Six studies were conducted in community settings [11,13,14,24,25,27], one was conducted in a residential setting (old age home) [26], and one was conducted in community and institutionalized long-term care (LTC) facilities (nursing homes) [12]. Two studies were conducted in rural India [24,25], four in urban India [12,13,14,26], and two in both rural and urban India [11,27]. Seven studies recruited both male and female participants [11,13,14,24,25,26,27], and one recruited only female participants [12]. The sample size of the included studies varied from 55 to 240. The standard definition of falls was used, except in four studies that did not mention the definition [12,14,25,27]. However, there were variations in how falls were measured. For example, two studies collected data on the history of falls in the past 12 months [14,24], two studies in the past 6 months [13,26], and the remaining four studies did not report any such criteria [11,12,25,27]. Four studies mentioned the location of falls, such as inside or outside the home [12,13,14,25]. Three studies collected data on recurrent falls but with varying definitions [12,13,26]. Two studies used the definition of recurrent falls as ≥2 falls in the last six months [13,26] whereas the other study used the definition of recurrent falls as ≥2 falls in the last year [12]. Falls were self-reported by participants in six studies [12,13,14,24,26,27]. Falls were reported by family members in one study [11] and by carers in the other study [25]. In addition, physical examination was performed in three studies [11,13,14], and three studies used medical records [11,24,25]. The health consequences of falls were self-reported by participants [11,12,13,14,24,26,27] and by carers [25]. Two of these studies also used medical records [13,25], and one study performed a physical examination [14].

### 3.3. Methodological Quality of the Included Studies

Table 2 presents the critical appraisal scores for the included studies. The scores ranged from 56% (five “yes” responses) to 78% (seven “yes” responses), out of a total of nine questions. All the studies had an appropriate sample frame to address the target population, as all of them only included older adults in India who fell. All the studies reported an appropriate sampling of participants, except two in which it was unclear [12,25]. Six studies reported the sample size calculation, but these sample sizes did not seem to be adequate [11,13,14,24,26,27]. These studies were neither national surveys nor large enough (the sample size ranged from 55 to 240 in these six studies) to estimate the prevalence of health consequences of falls among older adults in India, and the assumptions made in the sample size estimation were questionable. It was unclear in two studies, as the sample-size-estimation-related information was missing [12,25]. The study participants and settings were described in detail in all the studies. The data analysis was conducted with sufficient coverage of the identified sample (i.e., older adults in India who fell). Only one study reported using a valid method for the identification of the condition (i.e., a standardized scale for disabilities after falls) [11]. Four studies reported using a standard and reliable way to measure the health consequences of falls in all the participants (i.e., reported details about the data collectors, such as their training and experience) [12,14,24,25]. In terms of appropriateness of the statistical analysis, none of the included studies directly reported the prevalence (and 95% CI) of health consequences of falls, as the primary aim of some of these studies was different from that of this systematic review. However, raw data reported in these studies were used to calculate the prevalence and 95% CI. Five studies reported high response rates (from 92.1% to 98%) [11,13,14,24,26], but the other three studies did not provide enough information [12,25,27].

### 3.4. Meta-Analysis

All the studies were included in the meta-analysis.

Prevalence of injuries after falls

The pooled prevalence of injuries among older adults in India who fell was 65.63% (95% CI [38.89, 87.96]) (Figure 2). The statistical heterogeneity was 98.6% (*p* < 0.05).

2.Prevalence of head and/or neck injuries after falls

The pooled prevalence of head and/or neck injuries among older adults in India who fell was 7.55% (95% CI [4.26, 11.62]) (Figure 3). The statistical heterogeneity was 62.26% (*p* < 0.05).

3.Prevalence of upper extremity injuries after falls

The pooled prevalence of upper extremity injuries among older adults in India who fell was 19.42% (95% CI [16.06, 23.02]) (Figure 4). The statistical heterogeneity was 0% (*p* > 0.05).

4.Prevalence of trunk injuries after falls

The pooled prevalence of trunk injuries among older adults in India who fell was 9.98% (95% CI [2.01, 22.47]) (Figure 5). The statistical heterogeneity was 90.64% (*p* < 0.05).

5.Prevalence of lower extremity injuries after falls

The pooled prevalence of lower extremity injuries among older adults in India who fell was 34.36% (95% CI [24.07, 45.44]) (Figure 6). The statistical heterogeneity was 88.12% (*p* < 0.05).

6.Prevalence of cuts, lacerations, grazes, bruises, and/or contusions after falls

The pooled prevalence of cuts, lacerations, abrasions, grazes, bruises, and/or contusions among older adults in India who fell was 37.95% (95% CI [22.15, 55.16]) (Figure 7). The statistical heterogeneity was 95.33% (*p* < 0.05).

7.Prevalence of fractures after falls

The pooled prevalence of fractures among older adults in India who fell was 12.50% (95% CI [7.65, 18.30]) (Figure 8). The statistical heterogeneity was 79.91% (*p* < 0.05).

8.Prevalence of dislocations and/or sprains after falls

The pooled prevalence of dislocations and/or sprains among older adults in India who fell was 14.31% (95% CI [6.03, 25.26]) (Figure 9). The statistical heterogeneity was 89% (*p* < 0.05).

9.Prevalence of loss of consciousness after falls

The pooled prevalence of loss of consciousness among older adults in India who fell was 5.96% (95% CI [0.75, 15.08]) (Figure 10). The statistical heterogeneity was 90.7% (*p* < 0.05).

10.Prevalence of disabilities after falls

The pooled prevalence of disabilities among older adults in India who fell was 10.79% (95% CI [7.16, 15.02]) (Figure 11). The statistical heterogeneity was 16.4% (*p* > 0.05).

11.Prevalence of hospital admissions after falls

The pooled prevalence of hospital admissions among older adults in India who fell was 19.68% (95% CI [15.54, 24.16]) (Figure 12). The statistical heterogeneity was 0% (*p* < 0.05).

## 4. Discussion

The pooled prevalence of a range of health consequences of falls among older adults in India, including head and/or neck injuries, upper extremity injuries, trunk injuries, lower extremity injuries, cuts, lacerations, abrasions, grazes, bruises, and/or contusions, fractures, dislocations and/or sprains, loss of consciousness, disabilities, and hospital admissions, were reported. The findings are in line with studies conducted in various countries [28,29,30,31]. The USA’s Centers for Disease Control and Prevention (CDC) reported head injuries, fractures, and hospitalizations as serious consequences of falls among older adults [28]. Similarly, fall-related injuries and fractures were reported in studies conducted in Switzerland [29] and Spain [30]. A systematic review reported falls as a major contributor to spinal cord injuries among older adults [31]. Literature reviews conducted in India have also reported similar findings [32,33]. For example, a national review reported that every year nearly 1.5 to 2 million older people are injured and 1 million succumb to death due to falls [33]. The actual prevalence of fall-related health consequences could be much higher, and there could be under-reporting of the health consequences of falls. The findings indicate the seriousness of the problem and the need for interventions to prevent falls and manage their health consequences (including rehabilitation). It has been recommended that older adults, depending on their needs, should be offered multifactorial interventions (i.e., interventions targeting more than one risk factor) to prevent falls [34,35,36,37].

To the best of our knowledge, this was the first systematic review and meta-analysis to synthesize the existing evidence on the prevalence of health consequences among older adults in India. A robust systematic review process was followed using JBI and PRISMA guidelines. The chances of missing relevant articles were minimal, as a comprehensive search for both published and unpublished studies without any date or language restrictions was conducted. The included studies were conducted in community and residential care settings, and hence, the pooled prevalence might be more representative of the actual population. The findings could be generalizable to neighboring nations due to similarities in population, setting, and context. The case definition of falls and individual health consequences of falls was not always clear and consistent across studies. However, the authors tried their best to pool together the prevalence. This case definition issue could have also led to double counting, leading to an overestimation of the pooled prevalence of the health consequences of falls. The prevalence was pooled among older adults in India who fell and not among older adults in India in general, and thus, more primary research needs to be conducted before attempting such a systematic review. Although hospital admissions and length of hospital stay were included in this review, some experts may have a different view and do not consider these as direct health consequences of falls. Two of the included studies were conducted more than two decades ago [11,12], and the prevalence reported in these studies could be different from the current prevalence due to changes in India’s population characteristics and socioeconomic context. A meta-analysis could not be conducted on several health consequences of falls, as either these were not reported in any study (e.g., psychological health consequences, health-related quality of life, length of hospital stay, and death) or these were reported in single studies (e.g., pain (34.24%) [12] and post-fall syndrome (2.50%) [27]). Thus, more primary research on this topic needs to be conducted. The included studies were mostly conducted in the northern states of India, and thus, primary studies need to be conducted in other parts of the country for a more complete picture. In addition, the methodological limitations of the included studies should be addressed in future studies, such as small sample sizes and unstandardized data collection processes and tools.

In conclusion, the pooled prevalence of a range of health consequences of falls among older adults in India was reported, including the high prevalence of injuries, and indicating the need for prioritizing and addressing the problem. At the same time, given the limitations of the available evidence and to strengthen the evidence base, high-quality observational prevalence studies on this topic should be conducted and reported, including on psychological health consequences, health-related quality of life, length of hospital stay, and death. 

## Figures and Tables

**Figure 1 geriatrics-08-00043-f001:**
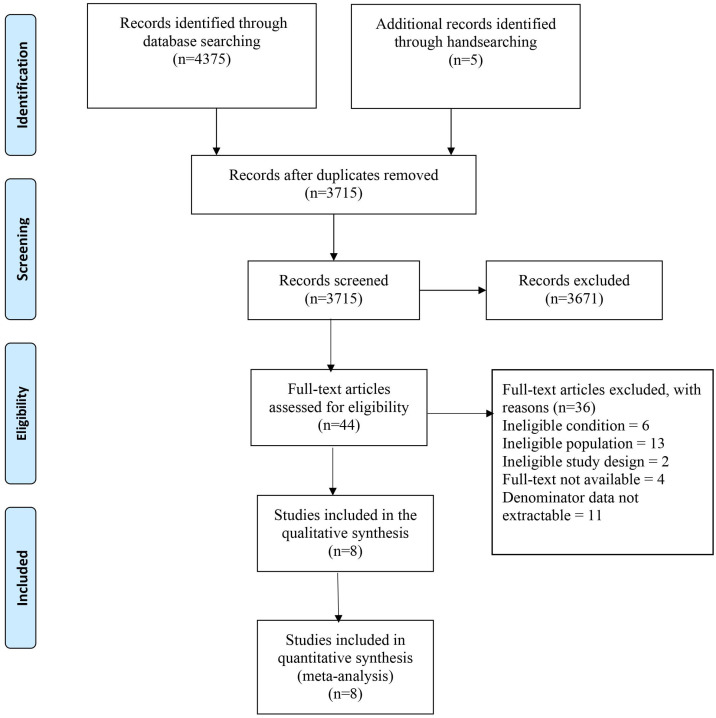
PRISMA flow diagram of the identification, screening, and eligibility of included articles.

**Figure 2 geriatrics-08-00043-f002:**
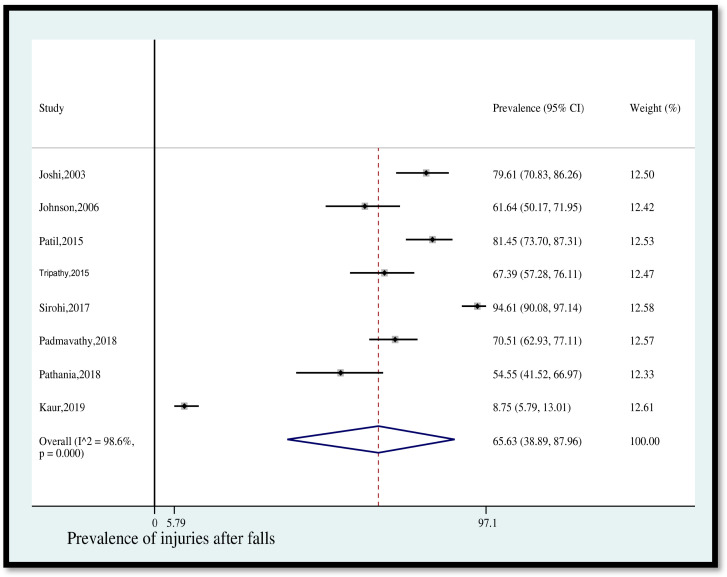
Pooled prevalence (in %) of injuries among older adults in India who fell. The dashed line represents the overall pooled prevalence. Joshi reported other fall-related injuries. Johnson separately reported leg/knee/ankle injury, hand/arm injury, head injury, back/neck injury, hip injury, and chest injury. Patil separately reported lower limb injuries, upper limb injuries, and reported head, back, and face injuries, altogether. Tripathy separately reported head/neck injury, trunk injury, spine injury, upper extremity injury, and lower extremity injury. Sirohi separately reported lower extremity injuries, upper extremity injuries, pelvis injuries, spine injuries, head injuries, and internal injuries. Padmavathy reported head and neck and upper extremity injury and lower extremity injury. Pathania separately reported upper limb injury, lower limb injury, head injury, spine injury, and neck injury. Kaur reported head injury and internal injuries. Refs. [11,12,13,14,24,25,26,27].

**Figure 3 geriatrics-08-00043-f003:**
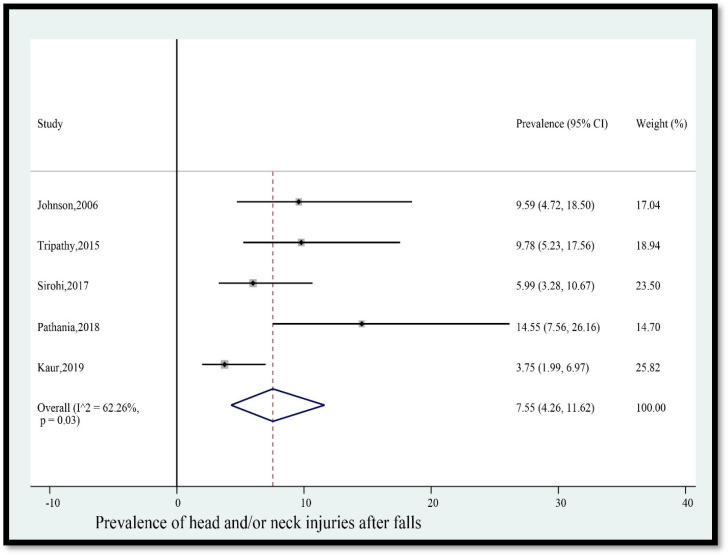
Pooled prevalence (in %) of head and/or neck injuries among older adults in India who fell. The dashed line represents the overall pooled prevalence. Johnson and Kaur reported head injury. Tripathy reported head/neck injury. Sirohi reported head injuries. Pathania separately reported head injury and neck injury. Refs. [12,14,24,26,27].

**Figure 4 geriatrics-08-00043-f004:**
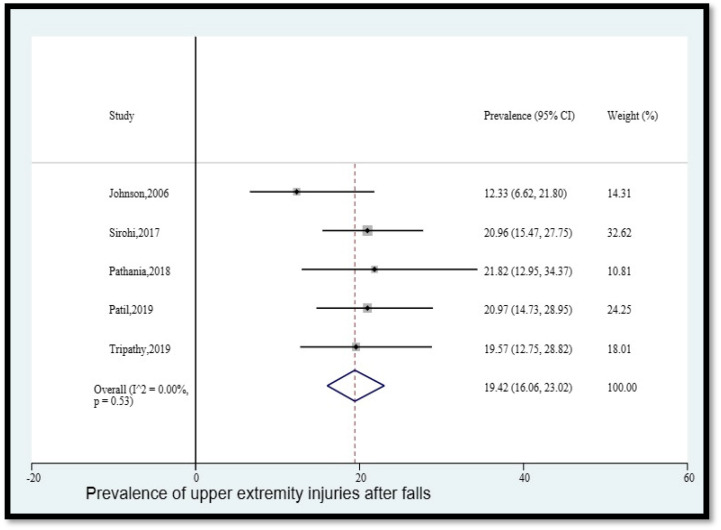
Pooled prevalence (in %) of upper extremity injuries among older adults in India who fell. The dashed line represents the overall pooled prevalence. Johnson reported hand/arm injuries. Sirohi reported upper extremity injuries. Pathania reported upper limb injury. Patil reported upper limb injuries. Tripathy reported upper extremity injury. Refs. [12,13,14,24,26].

**Figure 5 geriatrics-08-00043-f005:**
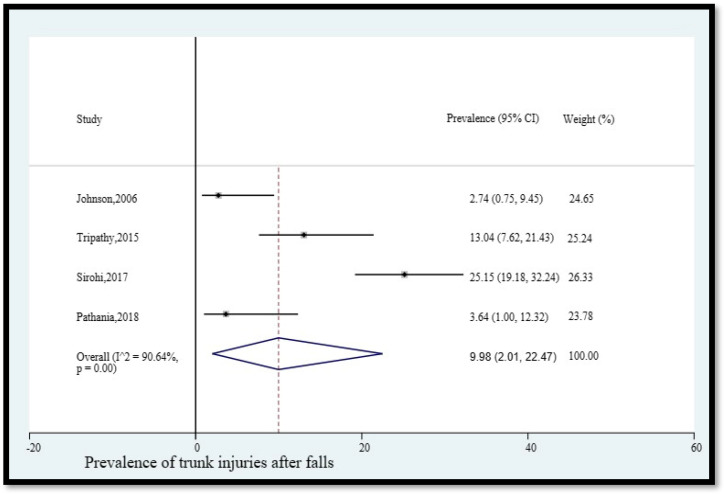
Pooled prevalence (in %) of trunk injuries among older adults in India who fell. The dashed line represents the overall pooled prevalence. Johnson reported chest injury. Tripathy separately reported trunk injury and spine injury. Sirohi separately reported pelvis injuries and spine injuries. Pathania reported spine injury. Refs. [12,14,24,26].

**Figure 6 geriatrics-08-00043-f006:**
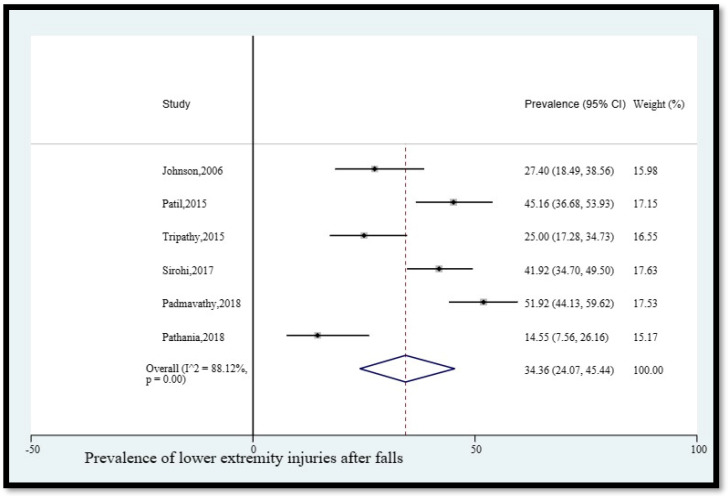
Pooled prevalence (in %) of lower extremity injuries among older adults in India who fell. The dashed line represents the overall pooled prevalence. Johnson separately reported leg/knee/ankle injury and hip injury. Patil reported lower limb injuries and Pathania reported lower limb injury. Tripathy reported lower extremity injury. Sirohi reported lower extremity injuries. Padmavathy reported lower extremity injury. Refs. [12,13,14,24,25,26].

**Figure 7 geriatrics-08-00043-f007:**
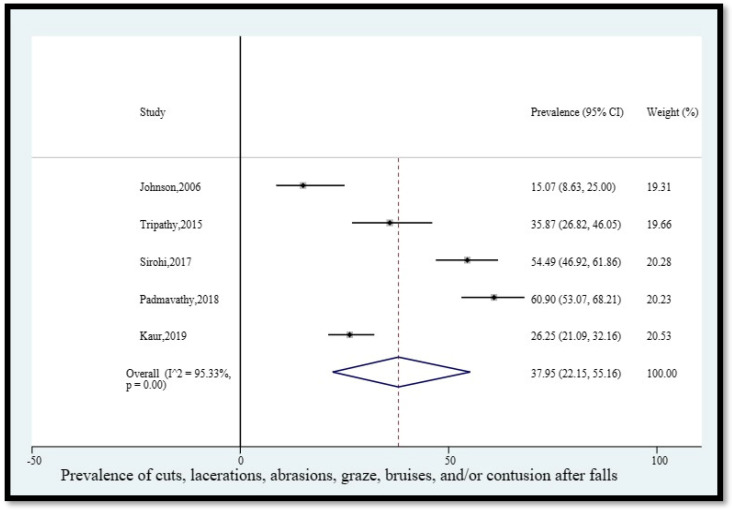
Pooled prevalence (in %) of cuts, lacerations, abrasions, grazes, bruises, and/or contusions among older adults in India who fell. The dashed line represents the overall pooled prevalence. Johnson reported minor cuts and bruises. Tripathy reported cut/bruise/abrasion. Sirohi reported cut/laceration/abrasion/bruise. Padmavathy reported contusion. Kaur reported bruises/cuts. Refs. [12,14,24,25,27].

**Figure 8 geriatrics-08-00043-f008:**
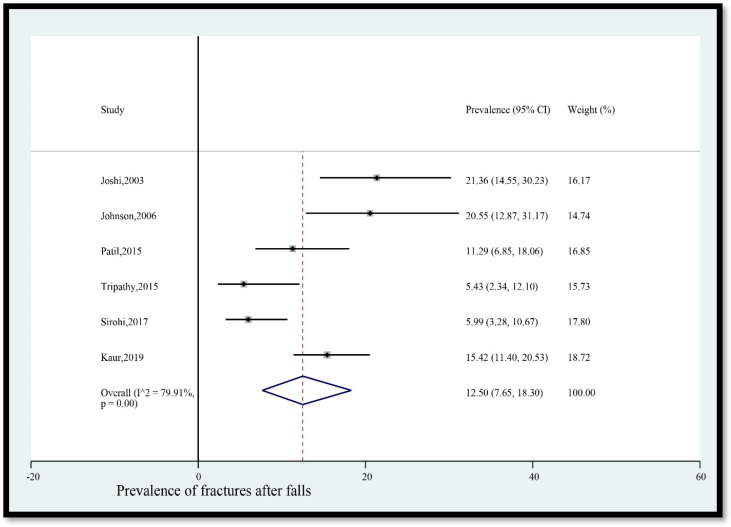
Pooled prevalence (in %) of fractures among older adults in India who fell. The dashed line represents the overall pooled prevalence. Joshi, Johnson, Patil, Sirohi, and Kaur reported fractures. Tripathy reported fracture. Refs. [11,12,13,14,24,27].

**Figure 9 geriatrics-08-00043-f009:**
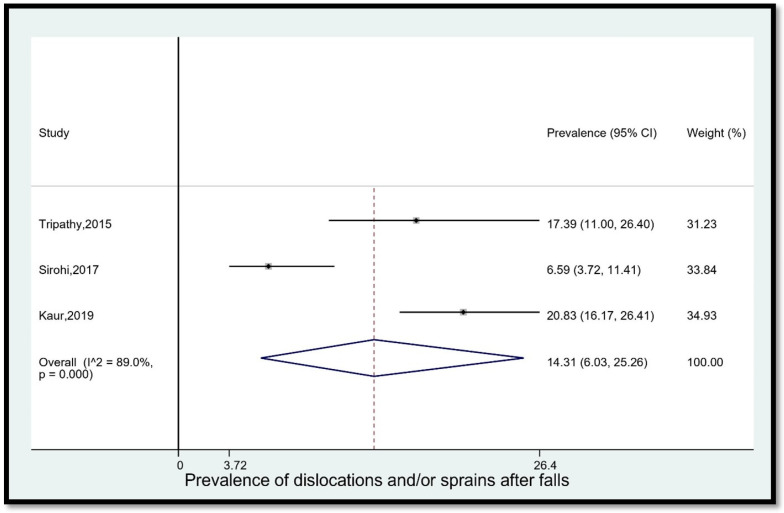
Pooled prevalence (in %) of dislocations and/or sprains among older adults in India who fell. The dashed line represents the overall pooled prevalence. Tripathy reported dislocation/sprain. Sirohi reported joint dislocation/sprain. Kaur separately reported sprains and dislocation of joints. Refs. [14,24,27].

**Figure 10 geriatrics-08-00043-f010:**
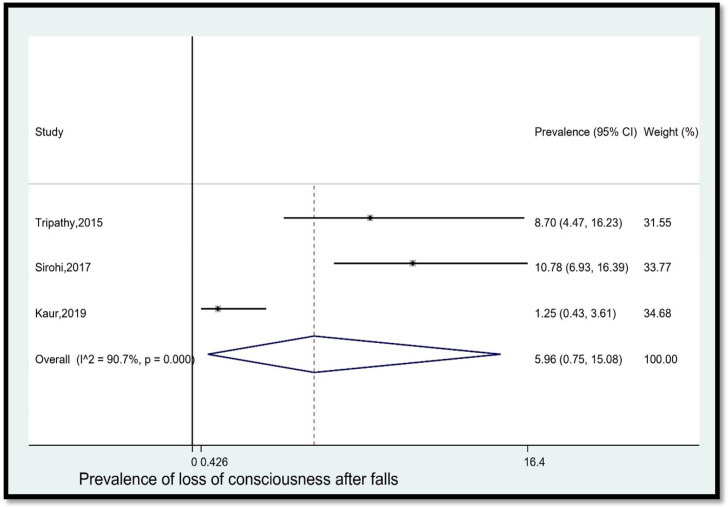
Pooled prevalence (in %) of loss of consciousness among older adults in India who fell. The dashed line represents the overall pooled prevalence. Refs. [14,24,27].

**Figure 11 geriatrics-08-00043-f011:**
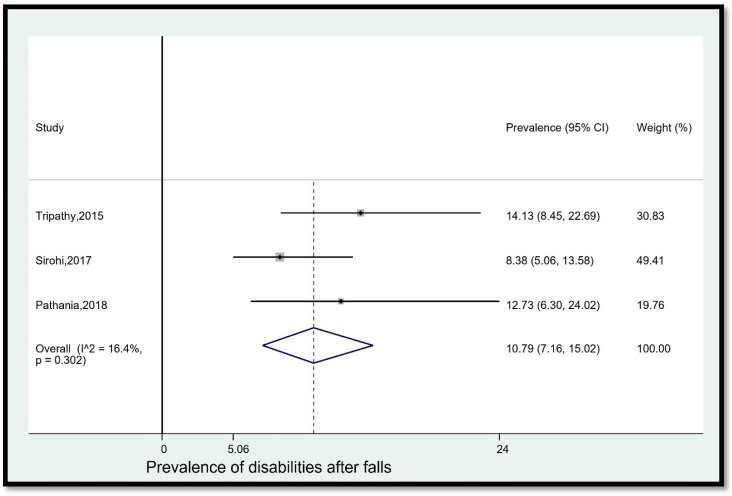
Pooled prevalence (in %) of disabilities among older adults in India who fell. The dashed line represents the overall pooled prevalence. Tripathy reported disability. Sirohi reported disability due to fall. Pathania reported disability after fall. Refs. [14,24,26].

**Figure 12 geriatrics-08-00043-f012:**
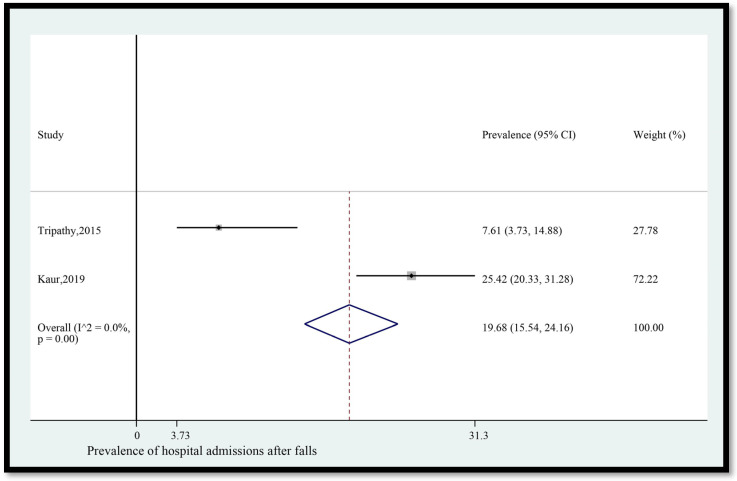
Pooled prevalence (in %) of hospital admissions among older adults in India who fell. The dashed line represents the overall pooled prevalence. Tripathy reported hospital admission. Kaur reported hospitalization. Refs. [14,27].

**Table 1 geriatrics-08-00043-t001:** Characteristics of included studies.

Author and Year	Study Design	Study Period	Indian State	Study Setting	Rural/Urban	Sex	Definition of Falls	Data Collection on Falls	Number of Falls	Data Collection on Health Consequences of Falls	Health Consequences of Falls and Numbers
Joshi, 2003 [11]	Cross- sectional	1999–2000	Haryana	Community	Rural and urban	Male and female	Ending up on the floor or ground unintentionally	Self-reported by family members, medical records, physical Examination	103	Self-reported by participants	Fractures = 21 Other fall-related injuries = 82
Johnson, 2006 [12]	Cross- sectional	2002	Kerala	Community and institutionalized long-term care (LTC) facilities (nursing homes)	Urban	Female	Not defined	Self-reported by participants	73	Self-reported by participants	Injuries sustained: pain = 25 fractures = 15 minor (cuts, bruises) = 11 Area of injury: head injury = 7 back/neck injury = 7 chest injury = 2 hand/arm injury = 9 hip injury = 3 leg/knee/ankle injury = 17
Patil, 2015 [13]	Cross- sectional	2010–2011	Karnataka	Community	Urban	Male and female	Inadvertently coming to rest on the ground, floor, or other lower level excluding intentional change in position to rest on furniture, wall, or other objects	Self-reported by participants and physical Examination	124	Self-reported by participants and medical records	Fall-related injuries = 101 Head/back/face injuries = 19 Upper limb injuries = 26 Lower limb injuries = 56 Bruises/internal injuries = 49 Sprains/grazes/cuts/others = 38 Fractures = 14
Tripathy, 2015 [14]	Cross- sectional	2011–2012	Punjab	Community	Urban	Male and female	Not defined	Self-reported by participants and physical examination	92	Self-reported by participants and physical examination	Fall-related injury = 62 Head/neck injury = 9 Trunk injury = 4 Upper extremity injury = 18 Spine injury = 8 Lower extremity injury = 23 Cut/bruise/abrasion = 33 Dislocation/sprain = 16 Fracture = 5 Loss of consciousness after falls = 8 Disability = 13 Hospital admission = 7
Sirohi, 2017 [24]	Cross- sectional	2015	Haryana	Community	Rural	Male and female	An event that results in a person coming to rest inadvertently on the ground or floor or other lower level	Self-reported by participants and medical records	167	Self-reported by participants	Head injuries = 10 Upper extremity injuries = 35 Spine injuries = 17 Pelvis injuries = 25 Lower extremity injuries = 70 Internal injuries = 1 Cut/laceration/abrasion/bruise = 91 Joint dislocation/sprain = 11 Fractures = 10 Loss of consciousness after fall = 18 Disability due to fall = 14
Padmavathy, 2018 [25]	Cross- sectional	2017	Puducherry	Community	Rural	Male and female	Not defined	Medical records and self-reported by carers	156	Medical records and self-reported by carers	Head and neck and upper extremity injury = 29 Lower extremity injury = 81 Contusion = 95
Pathania, 2018 [26]	Cross- sectional	2015	Delhi	Residential care (old age home)	Urban	Male and female	An event which resulted in a person coming to rest inadvertently on the ground or floor or other lower level	Self-reported by participants	55	Self-reported by participants	Fall-related injuries = 30 Head injury = 7 Neck injury = 1 Upper limb injury = 12 Spine injury = 2 Lower limb injury = 8 Disability after fall = 7
Kaur, 2019 [27]	Cross- sectional	2016–2017	Punjab	Community	Rural and urban	Male and female	Not defined	Self-reported by participants	240	Self-reported by participants	Head injury = 9 Internal injuries = 12 Bruises/cuts = 63 Dislocation of joints = 11 Sprains = 39 Fractures = 37 Loss of consciousness after fall = 3 Post-fall syndrome = 6 Hospitalization = 61

**Table 2 geriatrics-08-00043-t002:** Critical appraisal of included studies.

Author and Year	Q1	Q2	Q3	Q4	Q5	Q6	Q7	Q8	Q9	Total % of “Yes” to Critical Appraisal Questions
Joshi, 2003 [11]	Y	Y	N	Y	Y	Y	U	Y	Y	78
Johnson, 2006 [12]	Y	U	U	Y	Y	U	Y	Y	U	56
Patil, 2015 [13]	Y	Y	N	Y	Y	U	U	Y	Y	67
Tripathy, 2015 [14]	Y	Y	N	Y	Y	U	Y	Y	Y	78
Sirohi, 2017 [24]	Y	Y	N	Y	Y	U	Y	Y	Y	78
Padmavathy, 2018 [25]	Y	U	U	Y	Y	U	Y	Y	U	56
Pathania, 2018 [26]	Y	Y	N	Y	Y	U	U	Y	Y	67
Kaur, 2019 [27]	Y	Y	N	Y	Y	U	U	Y	U	56
Total % of “yes” to each critical appraisal question	100	75	0	100	100	13	50	100	63	

Y = yes, N = no, U = unclear. Q1. Was the sample frame appropriate to address the target population? Q2. Were study participants sampled in an appropriate way? Q3. Was the sample size adequate? Q4. Were the study subjects and the setting described in detail? Q5. Was the data analysis conducted with sufficient coverage of the identified sample? Q6. Were valid methods used for the identification of the condition? Q7. Was the condition measured in a standard, reliable way for all participants? Q8. Was there appropriate statistical analysis? Q9. Was the response rate adequate, and if not, was the low response rate managed appropriately?

## Data Availability

The data presented in this study are available within the article.

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
