# Peer review of "Health Consequences of Falls among Older Adults in India: A Systematic Review and Meta-Analysis"

_geriatrics, 2023, doi:10.3390/geriatrics8020043_

Round 1
Reviewer 1 Report
General comments
Thank you for the opportunity to review “Health consequences of falls among older adults in India: a systematic review and meta-analysis”. The authors present some interesting points throughout the manuscript and look to add to our understanding of falls epidemiology in India. However, there are some considerable limitations with this review which require major amendments. I believe that the review and meta-analysis adds to the evidence base sufficiently and with adequate amendments contributes to the academic community focusing on falls prevention and rehabilitation.
The protocol has been registered with PROSPERO and details have been reported.
Major comments
The searches were executed until 30th May 2022. Given the length of time passed since then, I suggest that searches are updated to potentially include more recent literature.
Lines 44-46: The mechanisms proposed for falls susceptibility need to be expanded upon. Given that understanding the causes of falls is so important to this topic, I suggest that the authors develop this section to provide a more detailed explanation. E.g. what physiological changes? Also, what is meant by weak postural control (line 45)? This is a very limited overview of a complex biomechanical age-related phenomena. Research has shown that joint powers are redistributed more proximally in older age, as well as reduced trunk and lower body joint ranges of motion, and diminished spatiotemporal gait parameters as conservative gait strategies to reduce falls risk. How then do older adults become more susceptible to falls because of weak postural control? Requires further explanation.
Introduction: The authors have not given a strong rationale for this review. The main rationale appears to be that a systematic review on this topic has not been conducted previously. The authors should consider providing a much more robust rationale. Furthermore, the aim does not appear to be commensurate with the methods. Did the authors aim to meta-analyse prevalence data on health consequences of falls? This should be clearer if so.
Whilst the authors assess the methodological quality of the included studies, there is very little interpretation of the risk of bias. Would the authors consider using the RoBANS tool or similar to assess the risk of bias? This is important as a study may be performed to the highest possible standards (i.e. high quality) yet still have an important risk of bias.
The methods used to investigate heterogeneity and explained variance in true effect should be reported. The authors should report their methods in more detail in line with PRISMA checklist – synthesis methods. The authors state that PRISMA was adhered to and should therefore make every effort to present the descriptions within the checklist.
Whilst I appreciate that the sub-group analyses contained smaller numbers of studies, I would like the authors to consider the use of sensitivity analyses to better understand their findings.
The meta-analyses figures are fairly well presented, however, reporting of the meta-analyses results is needed to support the figures.
Lines 315-317: The authors state that they omitted other important consequences of falls (e.g. hospital admission) as they were not a direct health consequence of the fall. I would argue that they are a direct consequence given that the fall would be the reason for their hospital admission. Including hospital admission and length of stay would greatly improve this review and meta-analysis as the authors could actually make more of an informed comment on the “seriousness of the problem” (line 294). Finally, can the authors comment on why they included such terms in the search strategy but decided to exclude from the data extraction/analysis?
Minor comments
Lines 10-13: The rationale for the research could be improved rather than being a series of related but fragmented sentences.
Line 14: JBI needs to be defined at first use.
Lines 31-33: The conclusion statement requires more support that is not apparent from the results in the abstract.
Lines 42-43: Whilst this is an abridged version of the definition for falls, ICD-10 codes include a range of falls (e.g. same level, upper level, and other unspecified falls). Would the author consider starting the introduction differently to reflect the range of falls older people experience?
Line 69: Would the authors consider changing “Inclusion criteria” to “Eligibility criteria”?
When reporting confidence intervals could the authors consider presenting it in APA format or similar (e.g. 95% CI [LB, HB])
Line 76: Can the authors justify why adults aged 60 years was chosen as the cut-off for inclusion?
Line 90: Given that personal pronouns have not been used throughout, would the authors consider removing the use of “we” here?
Would the authors consider removing “please see” when referring the reader to appendices?
Line 132: PRIMSA flow diagram – would the authors consider changing the top row right-hand box to “Additional records identified through handsearching”?
Characteristics of included studies section: Where the phrase “All studies… except … studies” has been used, can the authors consider rewriting to be clearer please? This section could be rewritten to be much clearer.
Lines 151 & 154: Would the authors consider changing caretakers to carers?
Lines 208-211: I’m not entirely sure what the authors are referring to here. Surely, the sample size is sufficient based on appropriate observed post-hoc power calculations? If not performed, and sufficient detail is not reported, then the author should state it wasn’t clear. Can the authors make it clearer what they are referring to here or amend as suggested?
Lines 216-218: What do the authors mean by “standard and reliable way”?
Lines 218-220: Is this necessary? It is very common that a review’s aims are different to the included articles. Also the reason studies did not include prevalence data as not because their aims were different to this review. Again, can the authors amend to make the meaning of the writing clearer please?
Lines 221-222: Can a range be provided for the response rates here please for comparison?
Could the authors include the unit used for prevalence in their meta-analyses figures? I assume it is % but needs to be explicit to avoid ambiguity.
Figure 2: The study by Kaur (2019) appears to be an outlier. Could the authors comment on this study in relation to the others please?
Figure 4: is the I2 value of 0% correct here?
Line 284: Other than when the authors have referred to a specific study, it is not clear what constitutes as a minor or major injury. Therefore, could the authors be more specific throughout the article regarding minor/major injury please? Also, if this was clearer, would the authors consider a separate sub-analyses on minor and major injuries? This could add an interesting discussion point.
Lines 288-289: I am unsure what the authors mean by global facts in the sentence “The findings are in line with global facts, emphasizing that falls can cause physical health consequences like injuries and fractures”.
Lines 294-295: I don’t believe the authors can conclude that the “findings indicate the seriousness of the problem and the need for fall prevention interventions”. The author’s have highlighted the prevalence of falls consequences, but not the consequence of falling (e.g. socioeconomic burden to healthcare).
Lines 317-320: Affected the findings in which way?
Lines 320-323: Can the authors provide references for these particular studies please? Whilst meta-analysing these studies may have been unachievable, could the results not have been qualitatively synthesised?
Reviewer 2 Report
This is a well written paper on an important topic. The authors clearly described their methods, and used appropriate methods to complete this systematic review. Table 1. is particularly well organized and clear.
I only have a few minor comments:
For the Figures (2-11): It is not clear what the dotted line represents, please describe this in the legend or the text.
On line 288, the authors write "The findings are in line with global facts..." It is unclear to me what exactly "global facts means", does it mean that there are similar prevalences by injury type in other countries? A brief summary of how the injury prevalences found by the authors compared to the injury prevalences in other countries, or groups of countries, or globally would provide important context and allow for greater interpretation of the significance of the findings.
Round 2
Reviewer 1 Report
Thank you for providing responses. Please see attached for further suggestions and comments.
